# Influences of Growth Stage and Ensiling Time on Fermentation Characteristics, Nitrite, and Bacterial Communities during Ensiling of Alfalfa

**DOI:** 10.3390/plants13010084

**Published:** 2023-12-27

**Authors:** Jiangbo An, Lin Sun, Mingjian Liu, Rui Dai, Gentu Ge, Zhijun Wang, Yushan Jia

**Affiliations:** 1Key Laboratory of Forage Cultivation, Processing and High Efficient Utilization, Ministry of Agriculture, Inner Mongolia Agricultural University, Hohhot 010019, China; an474608345@sina.com (J.A.); liumj_nm@163.com (M.L.); 18747659997@163.com (R.D.); gegentu@163.com (G.G.); zhijunwang321@126.com (Z.W.); 2Key Laboratory of Grassland Resources, Ministry of Education, Inner Mongolia Agricultural University, Hohhot 010019, China; 3College of Grassland, Resources and Environment, Inner Mongolia Agricultural University, Hohhot 010019, China; 4Inner Mongolia Academy of Agricultural & Animal Husbandry Sciences, Hohhot 010031, China; sunlin2013@126.com

**Keywords:** alfalfa silage, growth stage, nitrite, enzyme activity, bacterial communities

## Abstract

This study examined the impacts of growth stage and ensiling duration on the fermentation characteristics, nitrite content, and bacterial communities during the ensiling of alfalfa. Harvested alfalfa was divided into two groups: vegetative growth stage (VG) and late budding stage (LB). The fresh alfalfa underwent wilting until reaching approximately 65% moisture content, followed by natural fermentation. The experiment followed a completely randomized design, with samples collected after the wilting of alfalfa raw materials (MR) and on days 1, 3, 5, 7, 15, 30, and 60 of fermentation. The growth stage significantly influenced the chemical composition of alfalfa, with crude protein content being significantly higher in the vegetative growth stage alfalfa compared to that in the late budding stage (*p* < 0.05). Soluble carbohydrates, neutral detergent fiber, and acid detergent fiber content were significantly lower in the vegetative growth stage compared to the late budding stage (*p* < 0.05). Nitrite content, nitrate content, nitrite reductase activity, and nitrate reductase activity were all significantly higher in the vegetative growth stage compared to the late budding stage (*p* < 0.05). In terms of fermentation parameters, silage from the late budding stage exhibited superior characteristics compared to that from the vegetative growth stage. Compared to the alfalfa silage during the vegetative growth stage, the late budding stage group exhibited a higher lactate content and lower pH level. Notably, butyric acid was only detected in the silage from the vegetative growth stage group. Throughout the ensiling process, nitrite content, nitrate levels, nitrite reductase activity, and nitrate reductase activity decreased in both treatment groups. The dominant lactic acid bacteria differed between the two groups, with *Enterococcus* being predominant in vegetative growth stage alfalfa silage, and *Weissella* being predominant in late budding stage silage, transitioning to *Lactiplantibacillus* in the later stages of fermentation. On the 3rd day of silage fermentation, the vegetative growth stage group exhibited the highest abundance of *Enterococcus*, which subsequently decreased to its lowest level on the 15th day. Correlation analysis revealed that lactic acid bacteria, including *Limosilactobacillus*, *Levilactobacillus*, *Loigolactobacillus*, *Pediococcus*, *Lactiplantibacillus*, and *Weissella*, played a key role in nitrite and nitrate degradation in alfalfa silage. The presence of nitrite may be linked to *Erwinia*, unclassified_o__Enterobacterales, *Pantoea*, *Exiguobacterium*, *Enterobacter*, and *Allorhizobium–Neorhizobium–Pararhizobium–Rhizobium*.

## 1. Introduction

Alfalfa (*Medicago sativa* L.) is a perennial high–quality legume forage with a wide range of uses. It has the advantages of rich nutrient content, high biomass, strong stress resistance, and wide adaptability, and has extremely high feeding value and ecological value [1,2]. Alfalfa is widely planted in more than 80 countries, covering an area of more than 33 million hectares [3]. At present, alfalfa is mainly prepared as hay, but during the preparation process, due to factors such as rain, fallen leaves, and mildew, the nutritional components are greatly lost [4,5,6,7], while the actual production of silage is less affected by weather factors. Silage effectively preserves the nutrients found in alfalfa, ensuring their availability. The nutritional value of alfalfa silage is crucial for promoting animal growth and maintaining their health. This value is determined by the composition and proportion of essential nutrients such as proteins, carbohydrates, fats, vitamins, and minerals [8]. Likewise, the quality of silage fermentation plays a vital role in ensuring the safety and palatability of alfalfa silage. Factors such as microbial activity, acidity, flavor, and storage characteristics all contribute to fermentation quality [9,10]. During the same growing season, the duration of alfalfa’s growth surpasses that of other pastures within an identical timeframe, allowing for harvests 2–3 times or even 4–5 times per year [11]. Therefore, mowing at the growth stage when the nutritional value of forage is the highest and making silage can not only improve the quality of forage but also increase the number of mowing times in a year and increase the biomass of forage. Generally, leguminous forages have higher forage yield and nutritional value from the budding stage to the early flowering stage [12]. Soluble carbohydrates in pastures are the fermentation substrates of lactic acid bacteria, which are affected by the growth period of pastures and play an important role in the fermentation quality. Raguse et al. [13] found that the soluble carbohydrate content of alfalfa was significantly lower than that of red clover, the sucrose content of alfalfa was lower in the mature period than in the vegetative growth stage, and the sucrose content begins to decrease after the budding period.

However, due to the high water content, protein content, and buffer energy value of alfalfa, and the low content of water–soluble carbohydrates [14,15], the pH value drops slowly during silage, which not only leads to a decline in silage quality but also causes other bacteria to rapidly degrade. Breeding may lead to the accumulation of nitrate and nitrite in silage [16]. If ruminants eat a large amount of nitrate–containing feed, the amount of nitrate ingested and the reducing capacity of the rumen environment will be out of balance, and the nitrate will be reduced to nitrite and then absorbed into the blood, causing poisoning [17]. Under acidic conditions, nitrite reacts with some secondary amines in the feed or the stomach of the animal, and can also generate N–nitroso compounds (NOCs), including genotoxic and carcinogenic N–nitrosamines, which cause harm to ruminants [18,19]. The exposure and interconversion of nitrate, nitrite, and N–nitroso compounds can damage the central nervous system and skeletal system of animals, and even induce gene mutations in offspring, resulting in congenital deformities [20,21,22]. At present, the degradation of nitrite during silage is mainly related to the growth of lactic acid bacteria, while the accumulation of nitrite is more dependent on other microorganisms (such as *Escherichia coli*) [23]. Enterobacteriaceae and other enteric bacteria are harmful bacteria in silage, which can decompose sugars and proteins in silage, produce biogenic amines and branched chain fatty acids, reduce the quality of silage, and can also convert nitrate in silage into nitrite [24]. Bai et al. [16] and Zhao et al. [25] found that nitrate degradation in silage may be related to *Pantoea*, *Pseudomonas*, *Acinetobacter*, *Serratia*, *Aquabacterium*, *Enterobacter*, *Lactococcus*, and other bacteria.

Currently, a plethora of research exists concerning the role that nitrite plays in fermented products, such as fermented sausages [26], pickles [23], and cured meats [27]. However, there is relatively limited information available regarding the silage process. In recent times, researchers have gradually begun employing molecular techniques to analyze the variations in ensiling fermentation parameters and the dynamics of bacterial communities [28,29]. Nevertheless, previous studies have not thoroughly explored the specific microorganisms involved in the formation and degradation of nitrite during silage process. To the best of our knowledge, the complete characterization of the relationship between nitrate, nitrite, and their impact on fermentation quality, as well as the structure of microbial communities, has not been achieved in the context of alfalfa ensiling through natural fermentation. Hence, our hypothesis is that the diversity of microbial communities and nitrite content in alfalfa silage vary significantly at different growth stages. Moreover, there could be potential interactions between the microbial diversity within the alfalfa silage and the nitrite content during different growth stages, where certain microorganisms may either facilitate or obstruct nitrite formation. Furthermore, it is a well–established fact that there exists a close correlation between nitrite reductase and nitrate reductase activities, as well as nitrate and nitrite levels; however, studies exploring this relationship in silage are scarce.

Therefore, this study selected alfalfa in different growth stages (the vegetative growth stage and the late budding stage) for natural silage fermentation, aiming to explore the effects of different growth stages on the natural silage fermentation process of alfalfa, including the dynamics of nutritional quality, fermentation parameters, nitrate, nitrite content, nitrate reductase, nitrite reductase, and microbial community structure. Through conducting in–depth research on these transformations, we anticipate being able to furnish theoretical guidance and technical support for the production of alfalfa silage. Ultimately, our objective is to enhance both the quality and safety of alfalfa silage.

## 2. Results

### 2.1. Chemical Composition and Enzyme Activity of Alfalfa Raw Materials before Silage

According to Table 1, the chemical composition of alfalfa indicates significant differences between the vegetative growth stage (VG) and late budding stage (LB) treatment groups. Specifically, the CP content in the raw materials of the VG treatment group was notably higher than that of the LB treatment group (*p* < 0.05). Additionally, the WSC content, NDF content, and ADF content were significantly lower in the VG treatment group than in the LB treatment group (*p* < 0.05), while there was no significant difference in DM content between the two groups. However, it should be noted that the nitrite and nitrate content in the raw materials of the VG treatment group were significantly higher than those of the LB treatment group (*p* < 0.05), with levels of 18.70 mg/kg and 830.29 mg/kg, respectively. This was accompanied by a notable increase in both nitrite reductase and nitrate reductase activities in the raw materials of the VG treatment group compared to those in the LB treatment group (*p* < 0.05), which were 16.91 U/g and 15.40 U/g, respectively.

### 2.2. Chemical Composition and Fermentation Characteristics of Alfalfa Silage during Ensiling

As indicated in Table 2, the pH values, organic acid content, and NH_3_–N content of alfalfa silage were evaluated at different growth stages and fermentation characteristics. The interaction among these factors had a significant impact on the silage process (*p* < 0.05). With prolonged fermentation time, the pH values decreased, while the content of LA, AA, and PA increased. During fermentation, the VG treatment group displayed significantly higher pH values compared to the LB treatment group (*p* < 0.05). Specifically, the pH values of the VG treatment group decreased from 6.03 on day 1 to 5.06 on day 60, with significant declines noted on days 30 and 60 (*p* < 0.05). On the other hand, the pH values of the LB treatment group decreased from 5.53 on day 1 to 4.56 on day 60, exhibiting a significant decrease on day 3 (*p* < 0.05), followed by a noteworthy increase from day 15 to 30 (*p* < 0.05), and a subsequent significant decrease on day 60 (*p* < 0.05).

During the process of silage fermentation, there was a significant increase in the content of LA and AA in both the VG treatment group and the LB treatment group from day 1 to day 60 (*p* < 0.05). Specifically, in the VG treatment group, there was a significant increase in LA content on the 3rd day of silage (*p* < 0.05), followed by a significant decrease on the 5th day (*p* < 0.05). In the LB treatment group, LA content significantly increased on the 3rd day (*p* < 0.05) but remained stable from the 5th to the 10th day. From the 15th to the 30th day, LA content in both the VG and LB groups showed a significant increase (*p* < 0.05), while in the VG group, LA content significantly decreased at 60 days (*p* < 0.05). There was no significant difference in AA content between the VG treatment group and the LB treatment group on days 1, 3, and 7. However, starting from day 5 to day 60, the VG treatment group exhibited significantly higher AA content compared to the LB treatment group (*p* < 0.05). In the VG group, AA content significantly increased on the 3rd, 10th, and 30th days (*p* < 0.05), but decreased significantly on the 60th day (*p* < 0.05). In the LB treatment group, AA content significantly increased on days 5 and 7 (*p* < 0.05) and was significantly higher on day 60 compared to day 15 (*p* < 0.05). There was no significant difference in AA content from day 30 to day 60. Regarding PA content, there was a significant increase in the VG treatment group on days 7 to 15 (*p* < 0.05), followed by a significant decrease on day 60 (*p* < 0.05). There was no presence of PA in the LB treatment group on day 1, and the level of PA on day 10 was significantly greater than that on day 5 (*p* < 0.05). Furthermore, the PA content in the LB treatment group on the 30th day was significantly higher than that on the 15th day (*p* < 0.05), with no significant difference between the 30th and 60th days. The BA content in the VG treatment group showed a significant increase on the 7th and 10th days (*p* < 0.05), followed by a significant decrease on the 30th day (*p* < 0.05). No detectable presence of BA was observed in the LB treatment group. After 60 days of fermentation, the NH_3_–N levels in the VG treatment group and the LB treatment group were measured at 6.49 and 9.36, respectively. The lower NH_3_–N content in alfalfa silage was advantageous for enhancing its quality.

Table 3 displays the DM content, CP content, WSC content, NDF content, and ADF content of alfalfa silage at different growth stages on day 60. There were significant differences observed in these content parameters (DM, CP, WSC, NDF, and ADF) among the different growth stages of alfalfa silage (*p* < 0.05). Specifically, the VG treatment group exhibited significantly higher DM content, CP content, and WSC content compared to the LB treatment group on day 60. Conversely, the NDF content and ADF content were significantly lower in the VG treatment group compared to the LB treatment group (*p* < 0.05).

### 2.3. Nitrite Content, Nitrate Content, and Enzyme Activity during Alfalfa Silage

The dynamics of nitrite content, nitrate content, nitrite reductase activity, and nitrate reductase activity during the silage of alfalfa at different growth stages were examined in this study (Figure 1). Both the days of silage fermentation and the growth stage treatments, as well as their interaction, significantly influenced the nitrite content, nitrate content, nitrite reductase activity, and nitrate reductase activity during alfalfa silage (*p* < 0.05). The nitrite content of both treatment groups generally decreased with increasing fermentation time (Figure 1A). Specifically, the nitrite content of the VG treatment group significantly increased at 5 days of silage fermentation (*p* < 0.05) but then decreased from 5 to 30 days of fermentation. The nitrate content of the VG treatment group decreased significantly at 5 days of silage fermentation and further decreased at 60 days of fermentation (*p* < 0.05), being significantly lower than that at 30 days (*p* < 0.05) (Figure 1B). In the LB treatment group, the nitrite content showed a significant decreasing trend from 1 to 7 days of fermentation (*p* < 0.05), and was significantly higher at 30 days compared to 60 days of silage fermentation (*p* < 0.05) (Figure 1A). The nitrate content in the LB treatment group also showed a significant decreasing trend from 1 to 5 days and from 15 to 60 days of fermentation (*p* < 0.05) (Figure 1B). In terms of enzyme activity, the VG treatment group exhibited a significant decline in nitrite reductase activity from 3 to 15 days of ensiling (*p* < 0.05) (Figure 1C). The VG treatment group demonstrated significantly higher nitrate reductase activity at 5 days of ensiling compared to 3 days, followed by a significant decrease from 15 to 60 days of ensiling (*p* < 0.05) (Figure 1D). On the other hand, the LB treatment group showed a significant increase in nitrite reductase activity from 5 to 7 days of ensiling (Figure 1C). Nitrate reductase activity in the LB treatment group exhibited a significant decline from 1 to 7 days of ensiling (*p* < 0.05), with no significant difference observed from 15 to 60 days (Figure 1D).

### 2.4. Composition of Bacterial Community Structure during Alfalfa Silage Process

Table 4 displays the alpha diversity of bacteria during alfalfa silage. The duration of silage and growth stage significantly influence the Ace, Chao, Shannon, and Simpson indices of alfalfa silage (*p* < 0.05). Additionally, the interaction between silage fermentation days and growth period significantly impacts the Ace and Chao values during the process of alfalfa silage. Notably, the Ace and Chao indices of raw materials in the LB treatment group are significantly higher compared to the VG treatment group. However, no significant differences in the Ace and Chao indices were observed between the two treatment groups during the first 30 days of silage fermentation. After 60 days of fermentation, the Ace and Chao indices in the LB treatment group were significantly lower than those in the VG treatment group (*p* < 0.05). On the first day of alfalfa silage fermentation, the Simpson index in the LB treatment group was notably higher compared to the VG treatment group (*p* < 0.05). Moreover, as the fermentation time increased, Shannon showed a decreasing trend in both treatment groups, while Simpson exhibited an increasing trend.

The results of the prokaryotic microorganisms in different silage time samples are presented in a Venn diagram (Figure 2). As the silage time progressed, the overall abundance of special operational taxonomic units (OTUs) in the silage groups showed a gradual decrease, indicating the significant impact of silage days on the number of prokaryotic microorganism species. Among the 18 treatment groups, a total of 26 bacterial genera were identified. The LB treatment group had the highest number of unique bacterial genera in the raw materials (MR), with a total of 411, while the LB treatment group at 7 days had the lowest number of unique bacterial genera, with only 1.

The results of OTU–level principal coordinate analysis (PCoA) depicting the changes in bacterial community diversity within the two treatment groups are depicted in Figure 3. PC1 and PC2 accounted for 36.16% and 19.76% of the observed variations, respectively. Through these analyses, it was determined that the bacterial communities could be classified into six distinct categories. Specifically, the raw materials from both the VG and LB treatment groups exhibited similarities in terms of their assigned categories. The VG treatment group belonged to a single category on day 5, and from day 7 to day 30, it consistently remained within the same category. However, on day 60, the VG treatment group established its own uniquely defined category. Conversely, the LB treatment group formed its distinct cluster on day 1, and this categorization persisted from day 3 to day 60. These intriguing findings underscore the capacity of silage to effectively restore and reconfigure the bacterial community structure within alfalfa silage. In light of these outcomes, it is apparent that the intricate interplay between the treatment groups and the microbial communities orchestrates a dynamic reconstruction process.

The phylum–level composition of bacteria associated with alfalfa raw materials is presented in Figure 4A. The bacterial community attached to alfalfa raw materials primarily comprises Proteobacteria, Firmicutes, Bacteroidetes, Actinobacteria, and other bacterial phyla. Notably, the proportion of Proteobacteria exceeds 60% across all treatment groups, suggesting their dominant presence. Significant differences in bacterial composition at the phylum level were observed between the two treatments. In the alfalfa raw material of the VG treatment group, the predominant phyla were Proteobacteria, Firmicutes, and Actinobacteria, accounting for 60.67%, 34.08%, and 4.86%, respectively. Conversely, the bacteria attached to the alfalfa raw material of the LB treatment group mainly consisted of Proteobacteria, Actinobacteria, Firmicutes, and Bacteroidetes, representing 80.18%, 8.99%, 8.21%, and 1.94%, respectively.

The prevalent levels of fermentation bacteria in both groups of treatment were observed to be Firmicutes and Proteobacteria (Figure 4A). In the VG treatment group, on the initial day of silage fermentation, Firmicutes and Proteobacteria represented 41.64% and 52.89%, respectively. After 3 days, Firmicutes became the dominant bacteria, constituting 96.00%. As the silage was left to ferment for 5 days, the proportion of Firmicutes decreased while Proteobacteria increased. Within the VG treatment group, Firmicutes consistently remained above 96.74% from 7 days up to 30 days. However, after 60 days of silage fermentation, the VG treatment group experienced a decline in Firmicutes to 78.21% and an increase in Proteobacteria to 22.44%. In contrast, in the LB treatment group, on the first day of silage fermentation, Firmicutes and Proteobacteria accounted for 79.41% and 19.63%, respectively. Following a duration from 3 days up to 60 days, Firmicutes became the dominant bacteria, constituting 93.66% to 99.11%.

The composition of bacterial genera at the genus level in alfalfa raw materials, subjected to different treatments, is depicted in Figure 4B. Under the VG treatment, the bacterial genera identified include *Pandoraea*, *Pantoea*, *Exiguobacterium*, *Erwinia*, *Staphylococcus*, *Enterobacter*, *Pseudomonas*, *Sphingomonas*, *Allorhizobium–Neorhizobium–Pararhizobium–Rhizobium*, as well as other bacterial species. On the other hand, the bacterial genera observed in the LB treatment encompass *Weissella*, *Pandoraea*, *Enterobacter*, *Pantoea*, *Exiguobacterium*, *Enterococcus*, *Pseudomonas*, *Sphingomonas*, *Allorhizobium–Neorhizobium–Pararhizobium–Rhizobium*, and other species.

The composition of bacteria at the genus level during alfalfa silage at different growth stages is shown in Figure 4B. As the fermentation progressed in both treatment groups, the diversity of the microbial composition of the silage samples decreased. The fermentation process of the VG treatment group primarily consisted of *Enterococcus*, *Weissella*, *Lactiplantibacillus*, *Aerococcus*, *Lactococcus*, *Pediococcus*, *Pandoraea*, and other genera. From day 3 to day 30 of fermentation, *Enterococcus* became the predominant bacteria in the VG treatment group, consistently accounting for more than 58.08% of the population. The highest abundance was observed on day 3 whereas the lowest was on day 15. Moreover, with increasing fermentation time, *Weissella* exhibited an upward trend. By day 60 of silage, the microbial diversity of the VG treatment group had increased, and the abundance of *Enterococcus* (27.38%) and *Weissella* (3.00%) had decreased. However, *Lactiplantibacillus*, *Pediococcus*, *Pandoraea*, *Aerococcus*, *Lactococcus*, and *Loigolactobacillus* abundances had all increased to some extent. The fermentation process of the LB treatment group mainly comprised *Weissella*, *Enterococcus*, *Lactiplantibacillus*, *Pediococcus*, *Enterobacter*, *Loigolactobacillus*, *Levilactobacillus*, and other genera. *Weissella* and *Lactiplantibacillus* emerged as the prevailing microorganisms during the fermentation procedure of the LB treatment group. As the fermentation period progressed, the prevalence of *Weissella* witnessed a decline. On the first day of silage, *Weissella* achieved its peak abundance (67.23%), while on the 60th day of silage, it reached its lowest level (27.24%). Conversely, the abundance of *Lactiplantibacillus* increased with prolonged fermentation time. The prevalence of *Lactiplantibacillus* exhibited a nadir at the onset of fermentation (5.30%) and reached its zenith after 60 days (48.51%).

Based on the linear discriminant analysis (LefSe) applied to the relative abundance of silage bacteria (*p* < 0.05, LDA > 4.0, Figure 5), divergent microbial communities were observed in the forage and silage feed of both the VG treatment group and the LB treatment group. This disparity signifies discrepancies in the microbiota composition of the silage materials and feeds between the two treatment groups. The VG treatment group had *Exiguobacterium* and *Staphylococcus* as differential microbial groups in the raw materials, with *Exiguobacterium* still present after 1 day of ensiling. The LB treatment group had *Pseudomonas* and *Enterococcus* as differential microbial groups in the raw materials, with *Weissella* and *Pantoea* appearing as differential microbial groups after 1 day of ensiling. After 3 days of fermentation, the VG treatment group exhibited a distinct microbial population of *Enterococcus* and *Lactococcus*, whereas the LB treatment group showed a prevalence of *Weissella*. As the silage fermentation progressed from 5 days up to 60 days, both the VG and LB treatment groups displayed varying compositions of microbial species. The differential bacteria observed in the VG treatment group included *Enterococcus*, *Aerococcus, Pandoraea, Pediococcus*, and *Lactococcus*, while the LB treatment group exhibited *Weissella*, *Lactiplantibacillus*, *Pediococcus*, and *Levilactobacillus* as the dominant species.

### 2.5. Correlation Analysis between Nitrite Content, Nitrate Content, Fermentation Quality, and Microbial Community Structure in Alfalfa Silage

Figure 6B displays the correlation analysis between the content of nitrite and nitrate, and the fermentation quality of alfalfa silage. It was observed that the pH value and BA content exhibited a positive correlation with both nitrate and nitrite content. Conversely, LA content showed a negative correlation with nitrate and nitrite content. Additionally, AA content displayed a negative correlation with nitrite, while NH_3_–N content exhibited a negative correlation with nitrate content. We have conducted a more in–depth examination of the relationship between the nitrite and nitrate content in alfalfa silage and the microbial community structure, as illustrated in Figure 6A. There is a positive correlation observed between nitrites and *Erwinia*, unclassified_o__Enterobacterales, unclassified_f__Enterobacteriaceae, *Pantoea*, *Exiguobacterium*, *Enterobacter*, and *Enterococcus*. On the other hand, nitrites show a negative correlation with *Limosilactobacillus*, *Levilactobacillus*, *Loigolactobacillus*, *Pediococcus*, *Lactiplantibacillus*, and *Weissella*. As for nitrates, they exhibit a positive correlation with *Erwinia*, *Pantoea*, *Exiguobacterium*, *Lactococcus*, *Aerococcus*, and *Enterococcus*, while showing a negative correlation with *Limosilactobacillus*, *Levilactobacillus*, *Loigolactobacillus*, *Pediococcus*, *Lactiplantibacillus*, and *Weissella*.

The correlation between fermentation quality and bacterial communities was analyzed, as depicted in Figure 6C. The pH value showed a significant negative correlation (*p* < 0.05) with *Weissella*, *Limosilactobacillus*, unclassified_f__Lactobacillaceae, *Lactiplantibacillus*, *Levilactobacillus*, *Pediococcus*, and *Loigolactobacillus*, while it exhibited a significant positive correlation (*p* < 0.05) with *Enterococcus*, *Enterobacter*, *Pandoraea*, *Pantoea*, and *Erwinia*. LA content demonstrated a significant positive correlation with *Weissella*, *Lactiplantibacillus*, *Limosilactobacillus*, *Levilactobacillus*, *Pediococcus*, and *Loigolactobacillus*, and a significant negative correlation with *Enterococcus*, *Enterobacter*, unclassified_f__Enterobacteriaceae, *Exiguobacterium*, *Pantoea*, and *Erwinia*. AA content displayed a significant positive correlation with *Lactiplantibacillus*, *Levilactobacillus*, and *Pediococcus*, while it exhibited a negative correlation with *Enterobacter*, *Pantoea*, *Erwinia*, and unclassified_f__Enterobacteriaceae (*p* < 0.05). PA content showed a positive correlation with *Enterococcus*, *Aerococcus*, and unclassified_o__Lactobacillales. BA content demonstrated a positive correlation with *Enterococcus*, *Aerococcus*, and unclassified_o__Lactobacillales, but a negative correlation with *Weissella* and *Lactiplantibacillus*. NH_3_–N content showed a predominant negative correlation with *Enterococcus*, *Aerococcus*, and *Exiguobacterium*, while it exhibited a positive correlation with *Weissella*, *Lactiplantibacillus*, *Limosilactobacillus*, and *Levilactobacillus*.

### 2.6. Predicting the Pathways of Bacterial Communities at Three Levels

On Pathway level 1 (Figure 7A), the LB treatment group exhibits a higher proportion in the areas of metabolism, genetic information processing, and environmental information processing in comparison to the VG treatment group on the 60th day. Moving to Pathway level 2 (Figure 7B), the LB treatment group demonstrates a greater abundance in global and overview maps, carbohydrate metabolism, membrane transport, translation, nucleotide metabolism, replication and repair, lipid metabolism, folding, sorting and degradation, and drug resistance (antimicrobial) when compared to the VG treatment group. Transitioning to Pathway level 3 (Figure 7C), both treatment groups experienced improvement in metabolic pathways. Specifically, on the 60th day, the LB treatment group shows enhanced activity in the biosynthesis of secondary metabolites, biosynthesis of amino acids, ABC transporters, carbon metabolism, ribosome, starch and sucrose metabolism, purine metabolism, glycolysis/gluconeogenesis, phosphotransferase system (PTS), pyruvate metabolism, and amino sugar and nucleotide sugar metabolism compared to the VG treatment group.

## 3. Discussion

### 3.1. Chemical Composition of Alfalfa Raw Materials before Silage

The chemical composition of silage is widely recognized to be significantly influenced by the growth stage. In the VG treatment group, the raw materials exhibited lower levels of WSC content, NDF content, and ADF content, whereas the CP content was higher. This can be attributed to the vegetative growth stage of the alfalfa plant, characterized by its rapid growth and nutrient accumulation. Yuan et al. [30] posit that the optimal period for harvesting alfalfa is during the bud stage, and early harvested purple alfalfa exhibits lower fiber content but higher CP content. Delaying the harvest lowers the quality of purple alfalfa, which aligns with the findings of this study. During this phase, the plant synthesizes sugars, such as glucose, fructose, and sucrose, through the process of photosynthesis, while the biosynthesis of cellulose is comparatively limited [31,32,33]. As the alfalfa plants progress in their growth, the number of leaves increases, expanding the photosynthetic area and subsequently enhancing its efficiency. Consequently, the WSC content progressively rises. Toward the late budding stage, the plants must fortify their cell walls to facilitate the development of flowers, buds, or fruits. Accordingly, there is an increase in cellulose synthesis, leading to a relatively higher cellulose content [34]. Additionally, during the vegetative growth stage, protein synthesis is heightened, and more nutrients, including nitrate and nitrite, are absorbed from the soil. Plants uptake nitrate through their roots and convert it to nitrite using nitrate reductase. Nitrite, as a byproduct of nitrate reductase, is further reduced to N_2_O [35]. Under the catalytic effects of ammonia enzyme, N_2_O can be transformed into ammonia. Subsequently, ammonia undergoes amino acid synthesis with α–ketoglutarate to generate glutamic acid. Glutamic acid can be further metabolized into other amino acids, such as glutamine and arginine. These intricate processes and enzymes enable plants to efficiently utilize nitrate and nitrite, converting them into nitrogen sources essential for growth and development [36]. This also elucidates why the raw materials of the VG treatment group display relatively higher levels of nitrate content, nitrite content, nitrate reductase activity, and nitrite reductase activity.

### 3.2. Chemical Composition and Fermentation Characteristics of Alfalfa Silage during Ensiling

The WSC levels in the raw materials of the LB treatment group were higher compared to the VG treatment group, which can serve as a valuable fermentation substrate for lactic acid bacteria during the initial stages of fermentation. Sufficient fermentation substrate enables the swift multiplication of lactic acid bacteria, leading to an accumulation of lactic acid, consequently causing a rapid decrease in the pH value [37]. Consequently, during the silage fermentation process, the LB treatment group exhibited lower pH values and a higher LA content compared to the VG treatment group, implying a superior fermentation effect in the VG treatment group. Furthermore, the pH values in the VG treatment group remained within the range of 5.62–6.03 for 30 days before fermentation. The relatively elevated pH value can be attributed to the higher protein content in the VG treatment group, which hinders a rapid decrease in the pH value due to increased buffering capacity. Sun et al. [38] also noted the challenging nature of achieving satisfactory quality in alfalfa silage, citing factors such as low DM content, low WSC content, and high buffering capacity (BC). As the duration of silage time increased, the AA content in both treatment groups gradually augmented, consistent with prior studies [39]. Li et al. [39] theorized that acetic acid production in silage may be attributed to facultative heterofermentative bacteria, particularly certain lactobacilli strains. Acetic acid originates from sugar decomposition and fermentation by heterofermentative lactobacilli, Enterobacteriaceae, and Clostridia [40]. The PA content exhibited higher values in the VG treatment group compared to the LB treatment group. Additionally, no detectable levels of BA were observed in the LB treatment group, whereas a small content of BA was present in the VG treatment group. This disparity may be attributed to the interplay between *Enterococcus* and *Enterobacter* within the VG treatment group. *Enterococcus* and *Enterobacter* engage in metabolic processes, particularly the breakdown of substrates (typically carbohydrates like glucose) via the glycolytic pathway. Consequently, the substrate is metabolized into pyruvate and propionic acid [41,42]. Conversely, the presence of butyric acid serves as an indicator of efficient silage fermentation or secondary fermentation, which is contingent upon the content of lactic acid present in the silage. Furthermore, the proliferation of heterolactic acid bacteria and yeasts in silage contributes to an elevation in BA content [43]. NH_3_–N assumes significance as a crucial metric for evaluating silage fermentation quality. The NH_3_–N content in both the VG and LB treatment groups demonstrated an increasing trend followed by a subsequent decrease. Zi et al. [44] and Su et al. [45] have posited that undesirable microorganisms such as *Clostridium* and *Enterobacter* can decompose proteins and metabolize amino acids through the aminolysis of amino acids within proteins, thereby yielding metabolites like NH_3_–N. Lower content of NH_3_–N in silage signifies reduced protein loss and enhanced utilization of feed protein by ruminants [46].

The chemical composition of both treatment groups decreased after 60 days of silage compared to the raw materials. Nonetheless, alfalfa silage with the vegetable growth stage displayed a distinct advantage in terms of nutrient content. Moreover, alfalfa silage with the vegetable growth stage exhibited superior fiber digestibility. Specifically, after 60 days of silage, the VG treatment group demonstrated higher levels of DM, CP, and WSC. Conversely, the VG treatment group had lower levels of NDF and ADF compared to the LB treatment group. Notably, the nutritional quality of the raw material significantly influenced the nutrient composition of the silage. This study revealed that the nutrient content of the raw material was higher in the VG treatment group than in the LB treatment group, and this performance was sustained after 60 days of silage. As a result, it can be concluded that the raw material’s nutritional quality played a crucial role in shaping the nutrient composition of silage.

### 3.3. Nitrate and Nitrite Content and Enzyme Activity during Alfalfa Silage

The levels of both nitrate and nitrite were found to be higher in the VG treatment group compared to the LB treatment group. This observation suggests that the nitrate content in the silage material has a significant impact on the nitrite content after the silage process. In other words, higher nitrate content leads to increased production of nitrite. It can be inferred that the nitrate content in silage plays a role in the conversion of nitrate to nitrite. On the 5th day, the VG treatment group exhibited a significantly higher content of nitrites and a significantly lower content of nitrates (*p* < 0.05). In contrast, the LB treatment group displayed an overall decreasing trend in nitrite and nitrate levels, which could be attributed to the quality of silage fermentation and microbial activity. In the early fermentation stages, the LB treatment group predominantly underwent fermentation by *Weissella*, known for its superior acid production ability compared to *Enterococcus* in the VG treatment group. As a result, the growth of Enterobacteriaceae and nitrate–reducing bacteria was inhibited. Conversely, the VG treatment group demonstrated weaker acid production in the pre–storage phase, leading to unstable fermentation and a potential increase in Enterobacteriaceae and nitrate–reducing bacteria, thereby elevating nitrite levels. *Enterobacter* and nitrate–reducing bacteria typically exhibit nitrate reductase activity, whereby nitrate is converted into nitrite [47].

The reduction of nitrate is an anaerobic metabolic process that usually occurs under anaerobic or microaerobic conditions. In such an environment, nitrate–reducing bacteria produce nitrate reductase to facilitate nitrate reduction and utilize it as an intracellular electron acceptor. This process is further accompanied by the formation of nitrite reductase, which promotes the conversion of nitrite to nitrogen in the form of N_2_ or NH_3_ [48,49,50]. Therefore, during the pre–fermentation period of silage, both treatment groups exhibited high–nitrate–reductase activity and a decreasing trend in nitrite reductase activity and nitrate content. Throughout fermentation, the pH value and organic acids predominantly contributed to the reduction of nitrite content, while nitrite reductase also played a significant role in its decrease, with a synergistic effect observed between acids and enzymes [26]. Notably, nitrite reductase activity in the VG treatment group demonstrated a significant decreasing trend from 3 to 15 days, possibly due to the catalytic reduction of nitrite content depleting the enzyme. Interestingly, the LB treatment group showed an increasing trend in nitrite reductase activity from 5 to 7 days of fermentation, contrasting with the results from the VG treatment group. This discrepancy may arise from the diverse composition of substrates in the VG and LB treatment groups, resulting in distinct microbial communities and the potential presence of different nitrite reductase–producing bacteria, thereby leading to varying activities. Nitrite reductase and nitrate reductase activities in both treatment groups displayed a declining trend during the late stages of fermentation. This decline could be attributed to the gradual depletion of substrates as fermentation progresses and the subsequent shift of microbial energy towards the utilization of endogenous carbon sources [51]. During this phase, the metabolic activities of microorganisms may also diminish, resulting in decreased production and activity of these enzymes [52]. Additionally, the microbial demand for nitrogen sources may decrease as well, further contributing to weakened nitrite reductase and nitrate reductase activities.

### 3.4. Composition of Bacterial Community Structure during Alfalfa Silage

The coverage values for both treatment groups exceeded 0.99, indicating the extensive and thorough measurement of microbial diversity in the samples, and the sequencing depth was adequate for capturing the dynamics of the bacterial community [53]. α–Diversity enabled us to focus on the abundance and even distribution of species in the samples, revealing the degree of microbial community diversity. At 60 days of silage compared to day 1 of silage, the Ace, Chao, and Shannon indices exhibited significantly lower values, while the Simpson index showed a significant increase in both treatment groups. This can be attributed to the gradual reduction in the oxygen content and pH value during silage fermentation, resulting in a decline in aerobic and acid–tolerant bacteria, thus decreasing microbial diversity within the silage [54]. Furthermore, the lactic acid bacteria, through acid fermentation, also inhibited the growth of undesirable bacteria [55].

In this investigation, the predominant phylum found in alfalfa silage was Proteobacteria, which is consistent with previous research [56]. During the initial day of fermentation, the VG treatment group exhibited a dominance of Proteobacteria and Firmicutes [57]. From day 3 of the silage process onward, Firmicutes became the prevailing bacterial community. The increase in Proteobacteria after 5 days of fermentation could be attributed to the inadequate acid production capacity of *Enterococcus* fermentation and the slow decrease in the pH values, consequently leading to an elevated population of proteobacteria. In the case of the LB treatment group, Firmicutes dominated after one day of silage as the anaerobic conditions and acidic environment favored their growth. On the 60th day, the abundance of Firmicutes in the VG treatment group was lower compared to the LB treatment group, while the abundance of Proteobacteria was higher in the former. This disparity may be attributed to the stronger acidic environment in the LB treatment group during the early stages of fermentation, resulting in increased Firmicutes in that group and concurrent suppression of Proteobacteria [58].

The developmental phase of plants significantly influences microbial diversity and population size [59]. According to the findings by Thompson et al. [60], the release of certain nutrients from fully developed tissues plays a crucial role in microbial growth and the diversity of microbial communities. This disparity in epiphytic microbial communities across different growth stages could be attributed to this factor. At the taxonomic level of genus classification, the VG treatment group primarily exhibited dominance of *Enterococcus*, whereas the LB treatment group was characterized by the prevalence of *Weissella* and *Lactiplantibacillus.*

On the 1st day of fermentation of fermentation in the VG treatment group, the abundance of *Enterobacter*, *Pandoraea*, *Exiguobacterium*, *Pantoea*, and *Erwinia* increased, likely due to the presence of a modest amount of oxygen and ample substrate within the silage. On the 3rd day of fermentation, *Enterococcus* gradually acclimated to the environment and became dominant as a result of reduced O_2_ levels and a decrease in the pH value within the silage. Simultaneously, the bacteriostatic activity of lactic acid bacteria eliminated spoilage and pathogenic bacteria [61], aligning with previous research findings. Consequently, the abundance of *Enterobacter*, *Pandoraea*, *Exiguobacterium*, *Pantoea*, and *Erwinia* decreased. *Lactococcus*, capable of producing lactic acid, exhibited greater activity during the fermentation phase before storage. However, its abundance gradually declined as fermentation time progressed. During the fermentation process, the abundance of *Weissella* bacteria increases significantly within a span of 5 to 30 days. As the fermentation process advances to day 60, the lactic acid progressively accumulates, thereby reducing the pH values of the silage environment and depleting oxygen levels in the feed. Consequently, a shift towards anaerobic conditions takes place. Specifically, *Lactiplantibacillus*, *Pediococcus*, and *Loigolactobacillus*, being well–adapted to acidic and anaerobic environments, thrive and proliferate due to the provision of a more conducive growth habitat. This ultimately leads to a marked increase in their population abundance.

On the 1st day of fermentation, the group treated with LB displayed a prevalence of *Weissella*. However, starting from day 3, the population of *Weissella* began to diminish, while *Lactiplantibacillus* saw an increase. Previous studies have indicated that *Weissella* is a vital heterotrophic lactic acid bacterium, playing a crucial role in initiating early fermentation. As fermentation progresses, the pH values decrease, thereby inhibiting the growth of *Weissella* [62]. Consequently, the subsequent stages of fermentation are facilitated by highly acid–tolerant, homofermentative lactic acid bacteria, which elucidates the growing prevalence of *Lactiplantibacillus* and *Levilactobacillus* over time. Moreover, the abundance of *Pediococcus* exhibited a declining trend during the later stages of fermentation. *Enterobacter* are generally regarded as harmful microorganisms during the process of silage fermentation. The highest content of Enterobacteriaceae was observed on the first day of silage fermentation. As the pH values gradually decreased, the population of other lactic acid bacteria increased, indicating a favorable silage fermentation process.

The PCoA analysis revealed that the diversity of bacterial communities at the OTU level was significantly altered following silage fermentation, suggesting a reconstruction of the bacterial community structure in alfalfa silage. The initial bacterial communities of both treatment groups resembled taxonomic units. However, as the fermentation time progresses, the bacterial communities of the VG treatment group and the LB treatment group gradually diverge from the initial raw materials, giving rise to two distinct bacterial communities. This indicates that as the fermentation time increases, the bacterial communities of both treatment groups begin to stabilize. The observed phenomenon further highlights the progressive separation and establishment of distinct bacterial assemblages with time in the respective treatment groups.

The relative bacterial abundance in silage was assessed using linear discriminant analysis (LefSe). The results revealed elevated levels of *Exiguobacterium* and *Staphylococcus* when the VG treatment group was applied for 1 day. It is widely acknowledged that the presence of *Exiguobacterium* and *Staphylococcus* in silage can potentially impede its optimal quality. These bacteria possess the ability to utilize nutrients, including sugars and other essential compounds, present in the feed, thus depleting their nutritional value. Moreover, *Exiguobacterium* can convert lactic acid and WSC into acetic acid or other byproducts [63]. Additionally, Dunière et al. [64] proposed that *Exiguobacterium* can produce amino nitrogen, ammonia, and biogenic amines during metabolic processes, potentially rendering it detrimental. *Staphylococcus*, such as *Staphylococcus pseudintermedius* [65] and *Staphylococcus aureus* [66], both known pathogens, can consume fermentation substrates in silage, leading to the production of metabolites like ammonia and fatty acids that can increase the pH values of the feed. Maintaining an acid–base equilibrium during the silage process is crucial, as elevated pH values may facilitate the growth of unfavorable microorganisms, thereby compromising silage quality.

### 3.5. Correlation Analysis between Nitrite Content, Nitrate Content, Fermentation Quality, and Microbial Community Structure in Alfalfa Silage

The research findings suggest that changes in the pH value can be attributed to variations in the microbial communities attached to different growth stages of alfalfa. Consequently, this leads to disparities in the organic acid content observed in this study. The pH value of the alfalfa silage exhibited a significant negative correlation with *Weissella*, *Limosilactobacillus*, unclassified_f__Lactobacillaceae, *Lactiplantibacillus*, *Levilactobacillus*, *Pediococcus*, and *Loigolactobacillus*. Conversely, a significant positive correlation was observed between the pH value and *Enterococcus*, *Lactococcus*, *Enterobacter*, *Pandoraea*, *Pantoea*, and *Erwinia*. In addition, the pH values demonstrate an opposite trend to those of LA and AA. The low pH values and high lactic acid content in the silage feed across both treatment groups can be attributed to the presence of a lactic acid bacterial community. This community efficiently metabolizes soluble carbohydrates into lactic acid, thereby rapidly reducing the pH values of the silage feed over a short period. As the pH values decreased, the abundance of potentially harmful bacteria such as *Enterobacter*, *Pandoraea*, *Pantoea*, and *Erwinia* also decreased. Although *Enterococcus* and *Lactococcus* initiated lactic acid fermentation during the early stages of ensiling, these species are sensitive to low pH values and cannot survive in an acidic environment [67,68]. Acetic acid, on the other hand, originates from heterofermentative lactic acid bacteria, Enterobacteriaceae, and *Clostridia*, which decompose and ferment sugars [40]. The presence of butyric acid showed a negative correlation with *Weissella* and *Lactiplantibacillus*, while a positive correlation was observed with *Enterococcus*. In this experiment, butyric acid was only detected in the VG treatment group, possibly due to fermentation by *Enterococcus*. Previous research has demonstrated that *Enterococcus* can produce butyric acid through the catalysis of glutamate [69,70].

The pH value is positively correlated with the nitrate and nitrite content, while negatively correlated with the lactic acid content (Figure 4A). This indicates that, as the lactic acid content increases and the pH values decrease in ensiled forage, both nitrate and nitrite content decrease. It is commonly believed that the production of lactic acid and the decrease in the pH values are responsible for nitrite consumption [26]. Changes in the pH values can influence the composition and function of microbial communities, thus affecting the generation or consumption of nitrate and nitrite. In addition to lactic acid significantly affecting nitrite content, acetic acid also contributes to it, consistent with the findings of Xiao et al. [71]. Furthermore, the study conducted by Yang et al. [72] suggests that under acidic conditions, carboxyl groups (–COOH) in organic acids can provide hydrogen ions (H^+^), and excess H^+^ reacts with NO_2_^−^ to produce NO and NO_2_, thereby degrading nitrites. In this study, nitrate content is negatively correlated with NH_3_–N, suggesting that good ensiling fermentation reduces nitrate degradation. Research has shown that higher pH values and NH_3_–N content in silage are associated with nitrate reduction [73]. Adding glucose, formic acid, or conducting wilting, tearing, and chopping have been proven to preserve more nitrate, while adding alkaline substances (urea, calcium carbonate) impairs the quality of silage and increases nitrate degradation [74]. Therefore, it is not surprising that all measures to improve the quality of silage would reduce nitrate degradation.

As widely acknowledged, lactic acid bacteria have the potential to decrease nitrite content in the course of fermentation [75]. This observation is consistent with the findings of the present study, which has identified the prominent involvement of *Limosilactobacillus*, *Levilactobacillus*, *Loigolactobacillus*, *Pediococcus*, *Lactiplantibacillus*, and *Weissella* in nitrite degradation. Prior research has suggested that lactic acid bacteria play an enzymatic role in nitrite degradation, while the dominant role is still played by organic acids, such as lactic acid [76]. For instance, Yu et al. [27] employed *Lactobacillus curvatus* and *Pediococcus pentosaceus* as dual starter cultures for acid meat processing, successfully inhibiting the growth of coliform bacteria and decreasing levels of nitrite and biogenic amines, thereby enhancing the quality and safety of acid meat. Similar advantages of microbial metabolism on nitrite reduction have been reported in cucumber fermentation studies, where the supplementation of *Lactobacillus plantarum* and *Pediococcus pentosaceus* led to favorable outcomes [77]. In this investigation, the generation of nitrites could potentially be linked to *Erwinia*, unclassified_o__Enterobacterales, *Pantoea*, *Exiguobacterium*, *Enterobacter*, and *Allorhizobium–Neorhizobium–Pararhizobium–Rhizobium*. The group of Proteobacteria known as *Allorhizobium–Neorhizobium–Pararhizobium–Rhizobium* has been discovered to enhance its biodegradation capabilities, specifically in the breakdown of benzoate and nitrogen metabolism, in the presence of challenging circumstances, as supported by research [78]. Reports suggest that members of *Exiguobacterium* possess the potential ability to survive as facultative anaerobes, either generating energy through aerobic respiration using oxygen, or through anaerobic respiration by using nitrate reductase to reduce nitrate to nitrite [79]. Another study on saline wastewater demonstrated that *Exiguobacterium* strains are capable of nitrogen removal through heterotrophic nitrification and aerobic denitrification, oxidizing ammonium to nitrite through heterotrophic nitrification and converting nitrite to nitrogen gas through aerobic denitrification [80]. Under circumstances characterized by insufficient oxygen, *Enterobacter* demonstrates its capacity to utilize nitrate and nitrite as electron acceptors, as well as sources of nitrogen for biosynthesis. *Enterobacter* employs nitrate reductase to catalyze the reduction of nitrogen compounds in nitrate, converting them into more reduced forms, such as nitrite. Furthermore, nitrite is further reduced into ammonium salts, known as ammoniacal nitrogen [50]. It has been reported that all members belonging to the Enterobacteriaceae family possess the ability to ferment glucose and respire nitrate [81]. *Erwinia* is deemed to bear a resemblance to conventional members of the Enterobacteriaceae lineage by virtue of their possession of these two aptitudes. Initial investigations carried out on *Erwinia* rendered remarkably divergent outcomes. The examination ascertained that among the 67 strains of *Erwinia* analyzed, 65 strains demonstrated the capacity to undergo glucose fermentation, resulting in acid production. Furthermore, all 65 strains exhibited proficiency in nitrate respiration, reducing it to nitrite [82]. *Pantoea*, being a constituent of the Enterobacteriaceae family, is commonly regarded as a detrimental microorganism in silage. As per Li et al. [83], *Pantoea* fulfills a role analogous to that of *Enterobacter* in silage. Nevertheless, Lv et al. [84] discovered that *Pantoea* has the capacity to degrade NH_3_–N. The decomposition of nitrates is markedly influenced (*p* < 0.05) by *Limosilactobacillus*, *Levilactobacillus*, *Loigolactobacillus*, *Pediococcus*, *Lactiplantibacillus*, and *Weissella*. In instances where the prevalence of these microbial communities is substantial, the level of nitrates diminishes. Therefore, it may be posited that these six bacterial genera serve as the principal agents responsible for the nitrate degradation within the alfalfa silage fermentation procedure. Nonetheless, additional verification is necessary to substantiate this supposition. In the research undertaken by Paik et al. [85], the focus lay on the assessability of nitrate reduction by four strains of lactic acid bacteria. The results of this inquiry suggest the absence of noteworthy nitrate reduction by the lactic acid bacteria. Nevertheless, it is worth highlighting that all strains subjected to testing still manifested nitrate degradation. Additional inquiry is exigent when it comes to the decomposition of nitrate by lactic acid bacteria in silage.

### 3.6. Predicting the Pathways of Bacterial Communities at Three Levels

Predicting fluctuations in bacterial populations can offer a more comprehensive comprehension of the underlying mechanisms that impact the dynamic alterations in the quality of silage. However, it should be acknowledged that this sort of analysis is rooted in the phylogenetics of 16S rRNA and does not directly measure metabolic pathways [28]. Hence, additional omics approaches, such as metabolomics and proteomics, are essential for delving deeper into the functional profiles of bacterial communities within silage fodder. In this study, PICRUSt2 expands upon the original PICRUSt1 technique to anticipate the functional potential of microbial communities using marker gene sequencing profiles [86]. These profiles were utilized to investigate the impact of the growth stage on the functional characteristics of bacterial communities in two distinct growth stages of silage.

On Pathway level 1, the LB treatment group displayed higher proportions in metabolism, genetic information, and environmental information processing, compared to the VG treatment group on the 60th day. Digging deeper into the Pathway level 2 analysis, the silage samples exhibited higher abundances in carbohydrate metabolism, nucleotide metabolism, and lipid metabolism. These findings highlight the robust metabolic capacity of the silage bacterial communities, particularly in carbohydrate metabolism, during the ensiling process. Remarkably, the LB treatment group demonstrated a relatively higher abundance in membrane transport, indicating its elevated microbial activity, which is advantageous for signal transduction and membrane transport within the bacterial community. Additionally, the LB treatment group showed higher abundances in folding, sorting and degradation, translation, replication, and repair pathways, suggesting that even after 60 days, the bacterial community in the LB treatment group was capable of rapid proliferation. Research conducted by Kilstrup et al. [87] and Martinussen et al. [88] has elucidated that microorganisms utilize nucleotides not only for replicating and synthesizing DNA and RNA but also for replicating and synthesizing their genetic material. This intricate process provides energy for diverse biological activities within the cell. Conversely, the VG treatment group exhibited lower drug resistance: and antimicrobial, indicating potentially diminished cellular vitality, community structure, and adaptability of microorganisms in this particular group. On Pathway level 3, both treatment groups displayed an augmentation in the metabolic pathways, signifying a sustained and robust metabolic activity of the bacterial community in the silage after 60 days. Throughout the process of fermentation, the LB treatment group exhibited higher abundances of metabolic pathways, biosynthesis of secondary metabolites, carbon metabolism, purine metabolism, and ABC transporters compared to the VG treatment group, while showcasing a reduced abundance of the two–component system. This suggests that a well–executed fermentation process possesses the capability to impede the transportation of detrimental bacteria within the membrane system by promoting the synthesis of metabolites in the silage, thereby facilitating the proliferation of lactic acid bacteria strains and lowering the pH values [89]. Importantly, this study unveiled that the most substantial proportions were observed in the glycolysis/gluconeogenesis, starch and sucrose metabolism, pyruvate metabolism, and amino sugar and nucleotide sugar metabolism pathways. The stimulation of carbohydrate metabolism in alfalfa silage may be associated with the enhanced relative abundance of lactic acid bacteria. Kanehisa et al. [90] and Hisham et al. [91] postulated that the inclusion of lactic acid bacteria augments the utilization of available carbohydrates, resulting in the production of lactic acid, acetic acid, terpenes, and polyketides. Subsequently, these carbohydrates are transported into or out of the cells to facilitate bacterial cell division. Hence, the pyruvate metabolism in alfalfa silage may potentially contribute to the increased levels of lactic acid and acetic acid. amino acids, being essential components in plants, play a crucial role in promoting primary metabolism and the synthesis of plant proteins. Ribosomes, as the key players, convert the genetic code into sequences of amino acids and construct intricate protein polymers by connecting amino acid monomers. During the fermentation process, both treatments witnessed an increase in amino acid biosynthesis and ribosome abundance. Previously conducted studies have proposed that the level of amino acid metabolism may reflect the capacity of bacterial populations in silage to synthesize amino acids from scratch [58]. In a study conducted by Bao et al. [92], it was found that the relative abundance of amino acid metabolism increased in well–fermented silage, which aligns with the findings of our present study. However, in another study, Bai et al. [93] concluded that amino acid metabolism was predicted to be down–regulated in well–preserved silage. Although this phenomenon remains challenging to explain, the upregulation of amino acid metabolism may potentially be associated with the augmented lactic acid content in silage, as the formation of lactic acid involves processes like amino acid decarboxylation and arginine deamination [94].

## 4. Materials and Methods

### 4.1. Silage Preparation

In the experimental plot of Inner Mongolia Yihe Lvjin Agricultural Development Co., Ltd. (Chifeng, China), the second crop of alfalfa (WL319HQ), which was the main local planting variety in Aluhorqin Banner, Chifeng City, was divided into three plots as replicates. To assess its chemical composition, fermentation characteristics, enzyme activity, and bacterial community structure composition, cutting was performed at different stages of growth: vegetative growth stage (VG, June 15th) and late budding stage (LB, June 26th). The alfalfa was wilted until the moisture content reached approximately 65%, then chopped into 2–3 cm lengths and thoroughly mixed. The resulting material was packed into polyethylene vacuum packaging bags (30 × 40 cm), with each bag filled with 400 g. The bags were subjected to vacuum treatment, and this process was repeated three times for each group. Prepare a total of 48 bags (2 growth stages × 3 replicates × 5 ensiling days) and store at room temperature (20–30 °C). Samples were taken after alfalfa wilting (MR) and at different intervals after silage (1 d, 3 d, 5 d, 7 d, 10 d, 15 d, 30 d, 60 d) for analysis.

### 4.2. Chemical Component and Fermentation Characteristics Analyses

At various time intervals (1, 3, 5, 7, 10, 15, 30, and 60 days), the silage bags were opened, and 10 g of silage sample was taken and placed into a beating bag. Then, 90 mL of sterile water was added, and the mixture was homogenized using a sterile clamshell homogenizer. The resulting silage extract was filtered through four layers of medical gauze to prepare the silage extract. A portion of the filtrate was immediately used to measure the pH values of the silage using an electrode pH meter (PHS–3C, INESA Scientific Instrument Co., Ltd., Shanghai, China) [37]. The ammonia nitrogen content was determined using the phenol–sodium hypochlorite colorimetric method [95]. The organic acid content, nitrate content, and nitrite content were analyzed using high–performance liquid chromatography [16,96]. The total activity of nitrate reductase and nitrite reductase was assessed using the Nitrate Reductase Activity Assay Kit (AKNM001M, Boxbio, Beijing, China) and the Nitrite Reductase Activity Assay Kit (AKNM004M, Boxbio, China).

The chemical composition after 60 days of silage was quantified in terms of its dry matter (DM). The DM content of both the raw materials and the silage samples was measured by subjecting them to a 72 h drying process in a controlled oven set at 65 °C [97]. Subsequently, the dried samples were finely ground using an FZ–102 factory sample mill (manufactured by Shanghai Yemao Equipment Co., Ltd., Shanghai, China) with a 1 mm sieve, enabling nutrient analysis. The nitrogen content was determined using the Kjeldahl method [98], and the resulting value was multiplied by 6.25 to calculate the crude protein (CP) content. The neutral detergent fiber (NDF) and acid detergent fiber (ADF) content were analyzed according to the Van Soest [99] method, employing an Ankom 2000 fiber analyzer (manufactured by Ankom, Macedonia, NY, USA). Additionally, a thermostable amylase was used, and the expression of ash was taken into consideration. Lastly, the anthrone colorimetric method was utilized to determine the content of water–soluble carbohydrates (WSC) [100].

### 4.3. Composition of Bacterial Community Structure

According to the manufacturer’s instructions, the genomic DNA of microorganisms was isolated from homogenized samples of alfalfa and silage by employing the FastDNA^®^ Spin Kit for Soil (MP Biomedicals, Shanghai, China). The quality and concentration of DNA were assessed using 1.0% agarose gel electrophoresis and the NanoDrop2000 spectrophotometer (Thermo Scientific, Shanghai, China), respectively. The V3–V4 variable region of the 16S rRNA gene was amplified through PCR, utilizing the forward primer 338F (5′–ACTCCTACGGGAGGCAGCAG–3′) and the reverse primer 806R (5′–GGACTACHVGGGTWTCTAAT–3′) [101], based on the extracted DNA’s template. The PCR reaction components comprised 5× *TransStart FastPfu* buffer (4 μL), 2.5mM dNTPs (2 μL), forward primer (5 μM) (0.8 μL), reverse primer (5 μM) (0.8 μL), *TransStart FastPfu* DNA polymerase (0.4 μL), template DNA (10 ng), and final volume adjusted to 20 μL. The PCR protocol entailed an initial denaturation at 95 °C for 3 min, followed by 27 cycles of denaturation at 95 °C for 30 s, annealing at 55 °C for 30 s, extension at 72 °C for 30 s, and a final extension at 72 °C for 10 min. The ABI GeneAmp^®^ 9700 was employed as the PCR instrument (Shanghai, China). The PCR products were retrieved using a 2% agarose gel and purified using the DNA gel recovery purification kit (PCR Clean–Up Kit, China Biotech, Beijing, China). The purity of the purified products was quantified by employing the Qubit 4.0 (Thermo Fisher Scientific, USA). Sequencing was accomplished on the Illumina PE300 platform (Shanghai Meiji Biomedical Technology Co., Ltd., Shanghai, China). The raw data were deposited in the NCBI SRA database (Accession No.: PRJNA1033347).

### 4.4. Statistical Analyses

The statistical analysis was conducted using SAS 9.4 software to perform one–way and two–way ANOVA at a significance level of *p* < 0.05. Alpha diversity analysis, such as Chao 1 and the Shannon index calculation, was carried out using the Mothur software (http://www.mothur.org/wiki/Calculators) accessed on 3 July 2023. Intergroup differences were assessed using the Wilcoxon rank–sum test. To evaluate the similarity of microbial community structure among samples, principal coordinate analysis (PCoA) based on the Bray–Curtis distance algorithm was employed. Additionally, PERMANOVA (permutational multivariate analysis of variance) was applied to determine the significance of differences in microbial community structure among sample groups. To identify significant bacterial taxa at the phylum to genus levels, linear discriminant analysis effect size (LefSe) analysis was conducted. The criteria for significance were set as LDA (linear discriminant analysis) score > 4 and *p* < 0.05. Correlation network analysis was performed to select species based on Spearman correlation with an absolute value of correlation coefficient (|r|) > 0.1 and *p* < 0.05. Bacterial community functional features were predicted using PICRUSt2 (version 2.2.0). The abundance of pathways at levels 1, 2, and 3 in the bacterial community was visualized using GraphPad Prism 10 software.

## 5. Conclusions

Alfalfa silage at the late budding stage showed relatively high lactic acid content and low pH value during the entire fermentation process, indicating that the fermentation characteristics of alfalfa silage at the late budding stage were better. The dominance of specific microorganisms responsible for fermentation differs between the vegetative growth stage and the late budding stage. The fermentation quality of lactic acid bacteria, with *Weissella* as the predominant species, outperforms that of *Enterococcus*. It is worth noting that the nitrite levels in both scenarios remained within safe limits. Moreover, higher levels of nitrate content in silage will induce the conversion of nitrate to nitrite. During the silage process of alfalfa, the lower pH value and higher lactate content inhibit the decomposition of nitrate and reduce the level of nitrite. Additionally, nitrite reductase and nitrate reductase enzymes play a crucial role in determining the levels of nitrate and nitrite. The key microorganisms involved in the breakdown of nitrite and nitrate in alfalfa silage are lactic acid bacteria potentially. Nitrite production may be linked to *Erwinia*, unclassified_o__Enterobacterales, *Pantoea*, *Exiguobacterium*, *Enterobacter*, and *Allorhizobium*–*Neorhizobium*–*Pararhizobium*–*Rhizobium.* This study provided insights into the impact of microbial community structure on nitrate and nitrite levels in silage. However, it did not explore the role of microbial metabolites in nitrite production, which presents a limitation. Therefore, future studies must employ additional omics methodologies, such as metabolomics and proteomics, to effectively investigate nitrite formation in silage.

## Figures and Tables

**Figure 1 plants-13-00084-f001:**
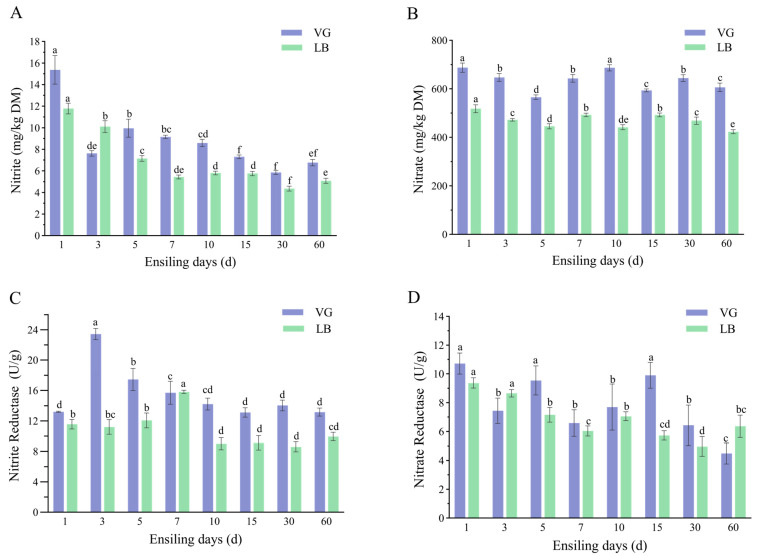
Nitrite content (**A**), nitrate content (**B**), nitrite reductase (**C**), and nitrate reductase enzyme activities (**D**) during alfalfa silage at different growth stages. Different lowercase letters (a–f) indicate significant differences in means (*p* < 0.05).

**Figure 2 plants-13-00084-f002:**
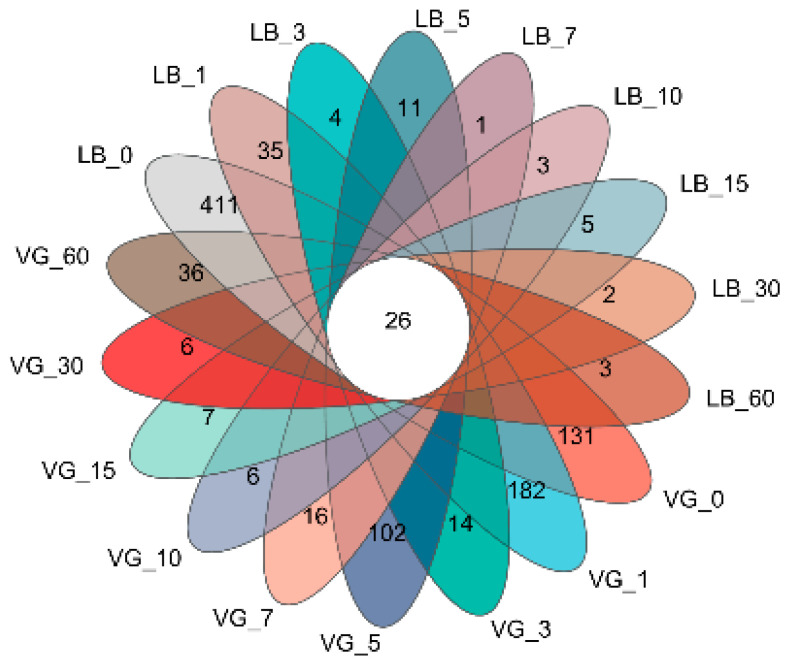
Number of genera and bacterial communities in alfalfa silage bacterial community.

**Figure 3 plants-13-00084-f003:**
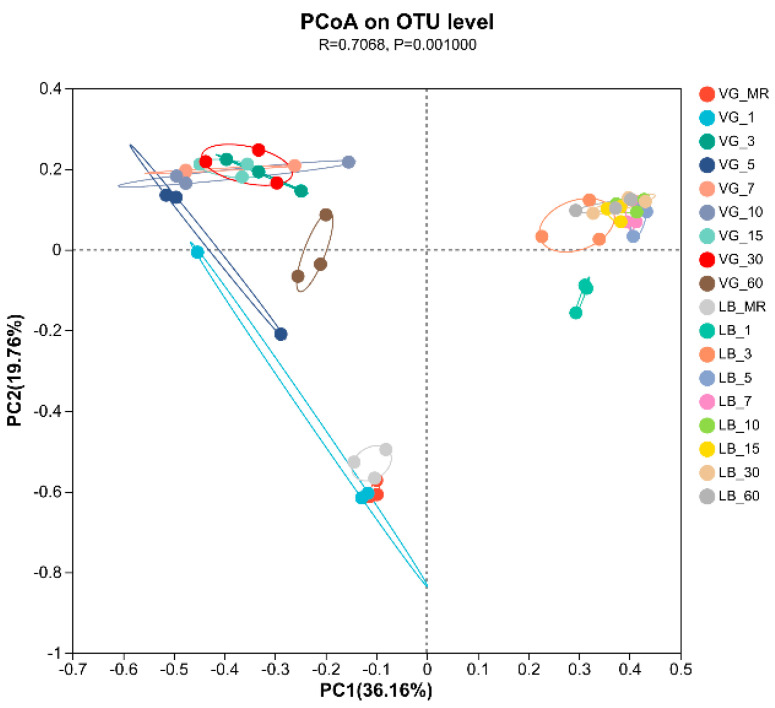
Principal component analysis (PCoA) of bacterial communities in silage raw material (MR) and alfalfa silage.

**Figure 4 plants-13-00084-f004:**
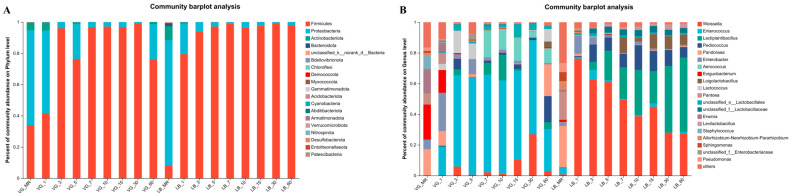
Relative abundance of bacterial communities at the phylum (**A**) and genus (**B**) levels in silage raw material (MR) and alfalfa silage.

**Figure 5 plants-13-00084-f005:**
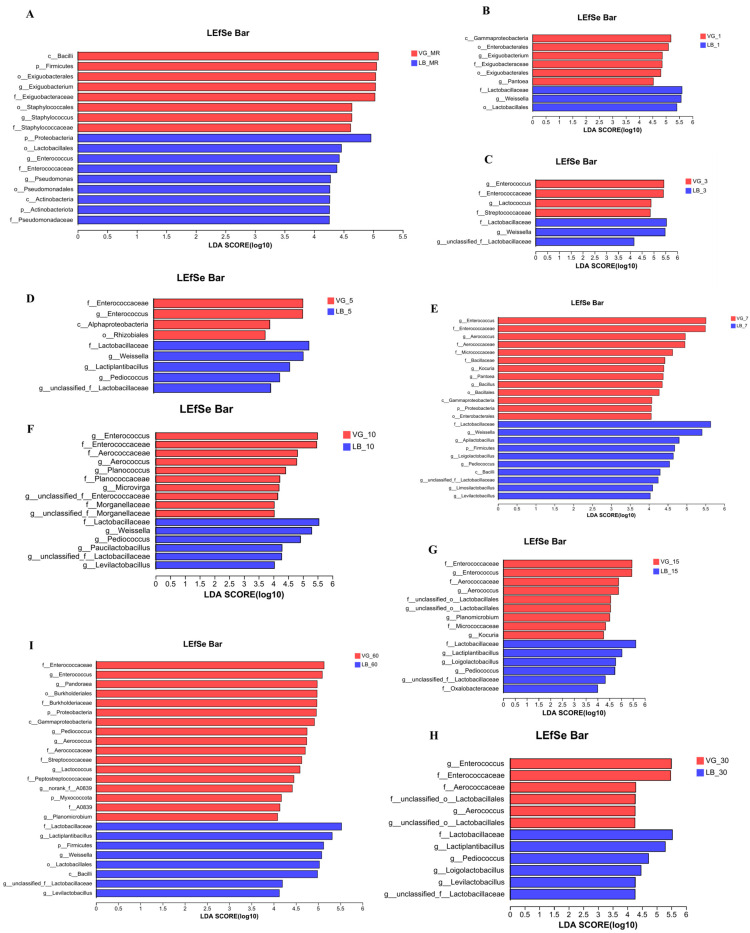
Compare the changes in microorganisms in alfalfa raw materials (**A**) and ensiling for 1, 3, 5, 7, 10, 30, and 60 days (**B**–**I**).

**Figure 6 plants-13-00084-f006:**
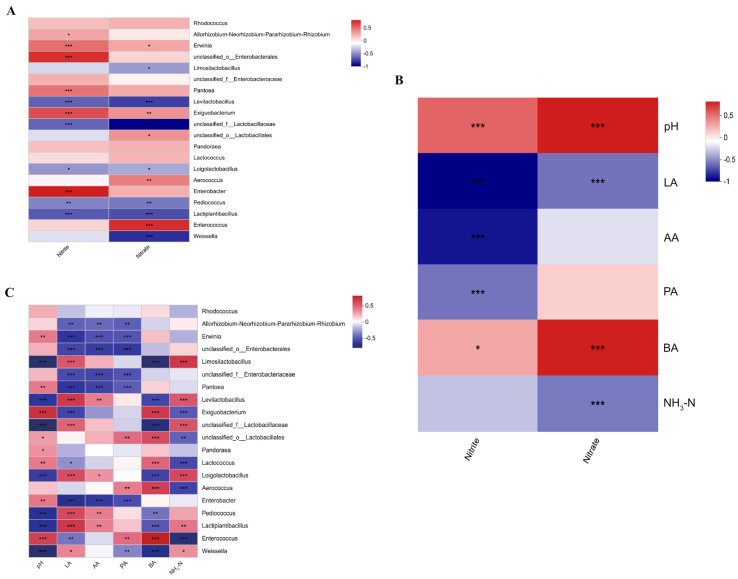
Correlation analysis between nitrite content, nitrate content, and microbial community structure (**A**) and fermentation characteristics (**B**) in alfalfa silage, as well as correlation analysis between fermentation quality and microbial community structure (**C**). LA, lactic acid; AA, acetic acid; PA, propionic acid; BA, butyric acid; NH_3_–N, ammonia nitrogen. *, 0.01 < *p* < 0.05; **, 0.001 < *p* < 0.01; ***, *p* < 0.001.

**Figure 7 plants-13-00084-f007:**
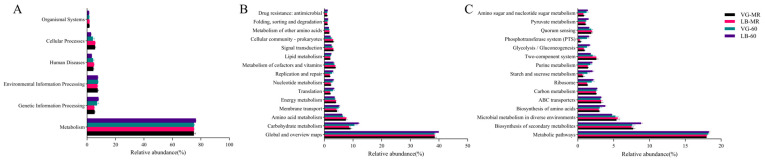
Predicted pathways of bacterial community in alfalfa raw material (VG–MR, LB–MR) and silage on day 60 (VG–60, LB–60). (**A**) Predicted pathways on level 1; (**B**) predicted pathways on level 2; (**C**) predicted pathways on level 3.

**Table 1 plants-13-00084-t001:** Chemical composition and enzyme activity of alfalfa silage raw materials.

Items	Growth Stages		SEM	*p* Value
	VG	LB		
DM (g/kg FM)	358.51 ^A^	353.89 ^A^	2.774	0.672
CP (g/kg DM)	29.70 ^A^	22.06 ^B^	0.42	0.007
WSC (g/kg DM)	24.93 ^B^	31.32 ^A^	0.522	0.004
NDF (g/kg DM)	264.27 ^B^	418.79 ^A^	8.503	0.011
ADF (g/kg DM)	244.92 ^B^	360.73 ^A^	10.325	0.017
Nitrite (mg/kg DM)	18.70 ^A^	15.45 ^B^	0.651	0.045
Nitrate (mg/kg DM)	830.29 ^A^	508.90 ^B^	15.343	0.008
Nitrite Reductase (U/g FM)	16.91 ^A^	12.53 ^B^	0.456	0.037
Nitrate Reductase (U/g FM)	15.40 ^A^	11.81 ^B^	0.526	0.021

DM, dry matter; FM, fresh matter; VG, vegetative growth stage; LB, late budding stage; WSC, water soluble carbohydrates; CP, crude protein; NDF, neutral detergent fiber; ADF, acid detergent fiber. Means with different letters in the same row (A–B) differ (*p* < 0.05). SEM, standard error of means.

**Table 2 plants-13-00084-t002:** Fermentation characteristics of alfalfa silage during ensiling.

Items	Growth	Ensiling Days								SEM	*p* Value		
	Stages	1	3	5	7	10	15	30	60		G	D	G × D
pH	VG	6.03 ^Aa^	5.87 ^Ba^	5.96 ^ABa^	5.89 ^Ba^	5.72 ^Ca^	5.84 ^Ba^	5.62 ^Ca^	5.06 ^Da^	0.032	<0.0001	<0.0001	<0.0001
	LB	5.53 ^Ab^	4.51 ^Eb^	4.60 ^DEb^	4.55 ^DEb^	4.62 ^Db^	4.73 ^Cb^	4.92 ^Bb^	4.56 ^DEb^				
LA	VG	13.89 ^Fb^	24.38 ^Db^	18.82 ^Eb^	24.88 ^Db^	26.51 ^Db^	31.42 ^Cb^	43.56 ^Ab^	38.62 ^Bb^	1.071	<0.0001	<0.0001	<0.0001
(g/kg DM)	LB	19.48 ^Ea^	29.76 ^Da^	35.02 ^Ca^	35.28 ^Ca^	35.95 ^BCa^	40.01 ^Ba^	57.75 ^Aa^	57.36 ^Aa^				
AA	VG	12.00 ^Ea^	14.54 ^Da^	16.48 ^Da^	16.72 ^Da^	19.72 ^Cb^	20.65 ^Ca^	32.35 ^Aa^	28.65 ^Ba^	0.692	<0.0001	<0.0001	<0.0001
(g/kg DM)	LB	9.66 ^Ea^	11.51 ^Ea^	15.54 ^Db^	20.04 ^Ca^	21.14 ^Ca^	21.86 ^BCb^	24.02 ^ABb^	25.32 ^Aa^				
PA	VG	1.22 ^Ea^	1.64 ^Ea^	2.41 ^Ea^	4.59 ^Da^	7.92 ^Ca^	8.56 ^BCa^	10.33 ^Ba^	6.89 ^Aa^	0.368	<0.0001	<0.0001	<0.0001
(g/kg DM)	LB	ND ^Db^	1.49 ^Ca^	1.69 ^Ca^	2.55 ^BCa^	3.58 ^Bb^	3.73 ^Bb^	5.90 ^Ab^	6.35 ^Aa^				
BA	VG	0.26 ^C^	0.33 ^C^	0.38 ^C^	0.84 ^B^	1.35 ^A^	1.41 ^A^	0.92 ^B^	1.05 ^B^	0.073	<0.0001	<0.0001	<0.0001
(g/kg DM)	LB	ND	ND	ND	ND	ND	ND	ND	ND				
NH_3_–N	VG	9.53 ^Aa^	8.23 ^Ba^	7.55 ^BCa^	7.74 ^BCb^	6.55 ^Db^	7.06 ^CDb^	6.44 ^Db^	6.49 ^Db^	0.322	<0.0001	<0.0001	<0.0001
(% TN)	LB	8.59 ^BCa^	8.66 ^BCa^	8.23 ^Ca^	11.10 ^Aa^	9.07 ^BCa^	9.01 ^BCa^	9.79 ^Ba^	9.36 ^BCa^				

DM, dry matter; TN, total nitrogen; VG, vegetative growth stage; LB, late budding stage; LA, lactic acid; AA, acetic acid; PA, propionic acid; BA, butyric acid; NH3–N, ammonia nitrogen. Means with different letters in the same row (A–E) or column (a–b) differ (*p* < 0.05). SEM, standard error of the mean; ND, no detected; G, the effect of growth stages; D, ensiling days; G × D, the interaction between growth stages and ensiling days.

**Table 3 plants-13-00084-t003:** Chemical composition of alfalfa silage on day 60.

Items	Growth Stages		SEM	*p* Value
	VG	LB		
DM (g/kg FM)	324.03 ^A^	310.67 ^B^	1.950	0.045
CP (g/kg DM)	29.93 ^A^	16.88 ^B^	0.553	0.007
WSC (g/kg DM)	15.12 ^A^	9.79 ^B^	0.628	0.004
NDF (g/kg DM)	318.44 ^B^	447.39 ^A^	4.463	0.005
ADF (g/kg DM)	282.08 ^B^	401.54 ^A^	3.051	0.001

DM, dry matter; FM, fresh matter; VG, vegetative growth stage; LB, late budding stage; WSC, water soluble carbohydrates; CP, crude protein; NDF, neutral detergent fiber; ADF, acid detergent fiber. Means with different letters in the same row (A–B) differ (*p* < 0.05). SEM, standard error of means.

**Table 4 plants-13-00084-t004:** Alpha diversity of alfalfa silage bacterial diversity.

Items	Growth Stages	Ensiling Days									SEM	*p* Value		
		0	1	3	5	7	10	15	30	60		G	D	G × D
Ace	VG	537.12 ^ABb^	626.79 ^Aa^	217.20 ^CDa^	365.66 ^BCa^	223.24 ^CDa^	213.57 ^CDa^	262.62 ^CDa^	137.42 ^Da^	369.04 ^BCa^	34.243	0.018	<0.0001	0.003
	LB	759.91 ^Aa^	407.25 ^Ba^	208.23 ^CDa^	248.00 ^Ca^	177.85 ^CDa^	173.57 ^CDa^	173.34 ^CDa^	132.58 ^Da^	175.73 ^CDb^				
Chao	VG	498.95 ^ABb^	564.49 ^Aa^	209.40 ^CDa^	355.95 ^BCa^	196.89 ^CDa^	199.09 ^CDa^	208.51 ^CDa^	122.66 ^Da^	271.64 ^CDa^	32.63	0.075	<0.0001	0.014
	LB	722.26 ^Aa^	389.51 ^Ba^	197.79 ^Ca^	204.84 ^Ca^	150.07 ^CDa^	139.26 ^CDa^	177.26 ^CDa^	117.78 ^Da^	152.75 ^CDb^				
Coverage	VG	0.9966 ^CDa^	0.9962 ^Da^	0.9987 ^ABa^	0.9977 ^BCa^	0.9987 ^ABa^	0.9988 ^ABa^	0.9987 ^ABa^	0.9992 ^Aa^	0.9979 ^ABb^	0.0003	0.006	<0.0001	0.219
	LB	0.9962 ^Ca^	0.9975 ^Ba^	0.9988 ^Aa^	0.9987 ^Aa^	0.9991 ^Aa^	0.9992 ^Aa^	0.9990 ^Aa^	0.9994 ^Aa^	0.9991 ^Aa^				
Shannon	VG	2.475 ^Aa^	2.442 ^Aa^	1.583 ^ABa^	1.572 ^ABa^	1.303 ^Ba^	1.116 ^Bb^	1.586 ^ABa^	1.333 ^Ba^	1.965 ^ABa^	0.213	0.006	0.001	0.098
	LB	3.19 ^Aa^	1.766 ^Ba^	1.886 ^Ba^	1.878 ^Ba^	1.880 ^Ba^	2.183 ^Ba^	2.027 ^Ba^	1.896 ^Ba^	1.896 ^Ba^				
Simpson	VG	0.195 ^Aa^	0.205 ^Ab^	0.414 ^Aa^	0.513 ^Aa^	0.485 ^Aa^	0.533 ^Aa^	0.393 ^Aa^	0.462 ^Aa^	0.210 ^Aa^	0.0716	0.006	0.268	0.070
	LB	0.165 ^Ba^	0.385 ^Aa^	0.308 ^ABa^	0.303 ^ABa^	0.206 ^ABa^	0.192 ^ABa^	0.240 ^ABa^	0.261 ^ABa^	0.306 ^ABa^				

VG, vegetative growth stage; LB, late budding stage. Means with different letters in the same row (A–D) or column (a,b) differ (*p* < 0.05). SEM, standard error of the mean; G, the effect of growth stages; D, ensiling days; G × D, the interaction between growth stages and ensiling days.

## Data Availability

Sequencing data for the 16S rRNA gene sequence were stored in NCBI with BioProject accession number PRJNA1033347.

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
