# Peer review of "Influences of Growth Stage and Ensiling Time on Fermentation Characteristics, Nitrite, and Bacterial Communities during Ensiling of Alfalfa"

_plants, 2023, doi:10.3390/plants13010084_

Round 1

Reviewer 1 Report

Comments and Suggestions for Authors

The study aimed to study the impacts of growth stage and ensiling duration on the fermentation characteristics, nitrite content, and bacterial communities during the ensiling of alfalfa.

The manuscript is within the scope of the journal and can be accepted after some revisions. The hypothesis of your manuscript and novelty should be detailed. After a quick literature search delivers a considerable number of publications that evaluated the effect of growth stage and ensiling duration on the fermentation characteristics, nitrite content, and bacterial communities during the ensiling of alfalfa. Therefore, there is little originality/novelty in the presented study in this aspect.

Authors should check all superscripts in Tables to match significance levels. For example, in Table 1: The P value is 0.045; however, both treatments carry "a" superscripts.

Author Response

Esteemed reviewer, I extend my sincerest gratitude for graciously dedicating a portion of your precious time to peruse this manuscript and for the invaluable insights you have provided. We have diligently addressed your comments and have diligently incorporated the necessary modifications, which are duly marked in the re-uploaded document. The following are the improvements we have made:

  1. In the introductory section, we have enhanced and refined the originality of our research and adjusted the underlying assumptions.
  2. We sincerely appreciate your astute observation concerning the error in Table 1's significance. We have rectified this discrepancy by carefully cross-checking the original data to ensure its accuracy within the table.

Reviewer 2 Report

Comments and Suggestions for Authors

Reviewer 3 Report

Comments and Suggestions for Authors

Dear Authors,

I commend you on the manuscript you submitted for publication to the Plants journal. The research you have described addresses a topic of utmost interest, and gaining a deeper understanding of how the microbial community modulates silage quality is crucial for enhancing its overall quality.

The methods are generally well-described. I ask you to provide more details on the bibliographic references regarding the silage method (lines 758 and 759).

The statistical approach is sound and does not require further elaboration.

The discussions are well-supported by the results and align with the manuscript's objectives.

I noticed some stylistic errors that need correction. I have indicated some in the revision, but I recommend an overall check of the formatting.

Therefore, I suggest that the paper can be accepted after minor revisions.

Comments on the Quality of English Language

The clarity of expression, precise language, and coherent flow make this text a pleasure to read.

I recommend only modifying lines 51 and 52 to make the text more fluid and avoid repetitions. I have provided a suggestion in the comment on the text.

So, minor editing of the English language is required.

Author Response

请参阅附件。

Round 2

Reviewer 2 Report

Comments and Suggestions for Authors
